# Mechanical Properties and Optimization Strategies of Tree Fork Structures

**DOI:** 10.3390/plants14020167

**Published:** 2025-01-09

**Authors:** Yi-Sen Peng, Bai-You Cheng, Tung-Chi Liu

**Affiliations:** 1Department of Horticulture, National Chung Hsing University, Taichung City 40227, Taiwan; d105032006@mail.nchu.edu.tw; 2Taiwan Wetland Society, Hsinchu County 30244, Taiwan; biyocheng@gmail.com

**Keywords:** tree fork, internal conical structure, branch–stem junctions, stress distribution, finite element analysis, mechanical properties

## Abstract

Trees are complex and dynamic living structures, where structural stability is essential for survival and for public safety in urban environments. Tree forks, as structural junctions, are key to tree integrity but are prone to failure under stress. The specific mechanical contributions of their internal conical structures remain largely unexplored. This study explores how conical structures optimize stress distribution, filling key gaps in the understanding of tree fork mechanics and supporting safety assessments. This study focused on the following factors: (1) external shape, (2) internal conical reinforcement, (3) interface of the conical connection, and (4) material changes within the conical structure. By analyzing physical samples to extract structural and morphological features, simulating these features in controlled variable models, and performing finite element analysis, we explored mechanical behavior, stress distribution, and performance characteristics, revealing the factors and mechanisms that strengthen tree forks. Insights from this study may facilitate safety assessments and inform pruning strategies for urban tree management.

## 1. Introduction

Trees, as living entities, rely on structural stability not only for their survival but also for public safety in urban green spaces. A vital element in this stability lies in tree forks, which form the connections between trunks and branches. Tree forks are key components of tree structure, highly susceptible to breakage, and serve as junctions for the vascular transport system connecting the trunk and branches [1]. Studies have highlighted the importance of in-depth investigations into the mechanical properties of tree forks for maintaining structural stability and supporting pathological defense [2,3]. Liu et al. explored the functional roles of various tree fork shapes and their mechanical properties through simulation analysis [4].

Tree forks exhibit not only external shape characteristics but also unique internal conical structures (Figure 1). Anatomical evidence and related studies suggest that the conical structure of a tree fork forms at the interface where the lateral branch connects to the trunk. The internal cone is formed by the trunk and the lateral branch through alternating secondary growth [2]. The annual rings of the lateral branches within the tree fork, along with the extended branch tail and stem tissue, grow in an alternate fashion. Because of differences in fiber orientation, the aforementioned components form a cross-layered structure of annual rings, which leads to the formation of an interpenetrating phase structure at the interface between the cone and the trunk. This arrangement facilitates efficient stress transfer and strengthens the connection between materials and structural components [5].

Although prior studies provide foundational insights, the internal conical structures’ role in stress mitigation and structural reinforcement remains poorly understood, representing a critical gap in tree biomechanics. Previous studies suggest that these structures can limit the spread of decay into the trunk when a branch is damaged or deteriorated [6,7,8]. In some species, such as pine, the internal cone may develop resinified heartwood, which further enhances material hardness and durability. Müller et al. [8] performed biomechanical analyses using polyester models and demonstrated the formation of cracks in branch–stem junctions under stress and the development of resinous wood. Garcia-Iruela et al. [9] and Nisula [10] reported significant mechanical differences between resinous and nonresinous wood.

Studies have highlighted the functional benefits of various external shapes of tree forks [2,3,4,10,11,12,13,14] and the disease resistance conferred by the conical structures within tree forks [6,12,13,14]. However, the specifics of how the internal conical structures meet the mechanical demands of branch–stem junctions remain unclear. The present study addressed this research gap by investigating the mechanical performance of the internal conical structure of tree forks, thereby offering a new theoretical framework for assessing tree structural safety. The objectives of this study were as follows:To investigate the effects of external shape and internal structure on the mechanical properties of tree forks;To analyze the mechanical properties of the connection between the trunk and the lateral branches through conical structures;To identify the mechanical benefits of conical interface interpenetration and cross-layer fiber reinforcement in strengthening the structure;To evaluate the effects of material changes, such as resinification and heartwood formation in cones and lateral branches, on the mechanical performance of tree forks.

By achieving these objectives, this study aims to reveal the mechanical strategies of tree forks in shape optimization, structural reinforcement, and material properties, providing scientific evidence for managing and protecting tree structural safety.

## 2. Results

To investigate the mechanical behavior of tree fork structures under load, we developed corresponding models and conducted finite element analysis (FEA). This section presents the FEA results pertaining to stress distribution and deformation characteristics, clarifying how the external shape and the internal structure influence the mechanical performance of tree forks.

### 2.1. Results of Stress Analyses

The results of stress analyses performed using finite elements (e.g., equivalent stress, shear stress, and total displacement) are presented in Figure 2. The analysis highlights how surface conditions and internal conditions influence the mechanical performance of tree forks across models A1 to C3.

Surface conditions refer to the external stress and shear stress distributions observed on the outer surface of the tree fork model.Internal conditions refer to the stress distributions (including von Mises stress and shear stress) and total displacement analyzed within the internal structure of the tree fork along the symmetry plane.

**Figure 2 plants-14-00167-f002:**
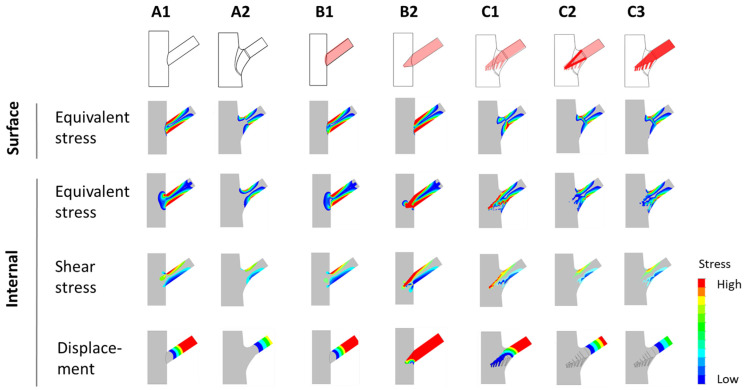
Diagrams depicting the distribution of stress and displacement. Stress and displacement distributions for various branch–stem junctions are presented. Surface stress and internal stress (equivalent and shear stress) are depicted alongside total displacement, with stress and displacement values shown through color gradients (e.g., red for high values and blue for low values).

### 2.2. Stress Curves

In addition to comparing models corresponding to the stress and total displacement diagrams (Figure 2), we analyzed stress–position data for key surface and internal areas; the results are presented in Figure 3. All results are arranged sequentially (A1, A2, B1, B2, C1, C2, and C3), covering the upper surface (abbreviated as “surface”), lower surface, and internal region. Data on stress, shear stress, and displacement were analyzed. The stress curve for the surface included both stress and shear stress, which for the lower surface included only stress and for the internal region included stress, shear stress, and total displacement. The contour length varied across the surface, lower surface, and internal region, resulting in different x-axis ranges: 0–180 mm for the surface, 0–230 mm for the lower surface, and 0–120 mm (stress and shear stress) and 0–200 mm (total displacement) for the internal region. The y-axis ranges were as follows: 0–2 MPa for the surface, 0–2 MPa for the lower surface, and 0–2 MPa (internal stress), 0–1 MPa (shear stress), and 0–0.2 mm (displacement) for the internal region.

### 2.3. Combined Curves

To better compare the mechanical performance across models, stress and displacement data from the surface, the lower surface, and the internal regions were integrated into combined curves (Figure 4, Figure 5 and Figure 6). This approach highlights the relationships between external and internal stress distributions, as well as the total displacement, providing a clearer understanding of the structural behavior under various conditions.

## 3. Discussion

### 3.1. Mechanical Analysis of Branch–Stem Junction Shape

#### 3.1.1. Effect of Shape on Surface-Stress Distribution

The stress analysis diagrams and the numerical results show that samples with similar shapes exhibit comparable surface-stress distributions. Specifically, the group without morphological features (A1, B1, B2) displayed similar patterns, while the group with features (A2, C1, C2, C3) also showed consistent surface-stress distributions. These findings indicate that the external shape of the tree fork plays a crucial role in influencing surface mechanical behavior.

#### 3.1.2. Mechanical Benefits of Tree Fork Morphological Features

By analyzing the stress distribution diagrams and stress–position curves of the feature-enhanced models (Figure 2 and Figure 4), it was observed that morphological features help disperse stress at the branch–stem junction and guide stress concentration toward specific locations. This mechanism shifts the failure point outward to the side branch, thereby protecting the main stem from structural damage.

### 3.2. Mechanical Analysis of Internal Connection Structures

#### 3.2.1. Stress Concentration at Connection Interfaces

A comparison between the monolithic models (A1, A2) and the other models revealed that two-part connected models exhibited significant stress concentration at the connection interface. This result indicates that the interface is a structurally vulnerable region (Figure 2 and Figure 5).

#### 3.2.2. Stress Concentration Guided by Conical Interlocking Structures

The internal connection structure, formed by the intergrown branch and stem, results in a conical interlocking configuration. This structure facilitates the transmission of external forces inward but also leads to stress concentration at the conical interface (Figure 2).

A comparison between the flat two-part connection (B1) and the conical interlocking connections (B2, C1) shows that the embedded interface in the conical models exhibits significant internal stress concentration (Figure 5), highlighting the vulnerability of the conical interface. In model C1, stress concentration was observed at the internal conical region (Figure 2, internal view). Müller [8,14] reported that some tree forks experience internal interface fractures, where ruptured cell vacuoles trigger the synthesis of secondary metabolites, resulting in the formation of occlusions and colloidal repair structures. These fracture locations align with the stress concentration regions in Figure 2 (C1), supporting the accuracy of the simulation results.

#### 3.2.3. Stress Redistribution Effects of Conical Structures

A comparison between the triangular conical model (B2), the branch conical model (C1), and the interface-reinforced model (C2) reveals the unique characteristics and advantages of the conical structure (Figure 2, internal view).

Figure 7 show the conical structure refers to the tapered transition zone formed at the connection between the branch and the main stem, typically consisting of a gradually expanding wood structure (Figure 7A,C,E). Its characteristics include the following:Stress redistribution through non-linear, stepped interfaces.The irregular stepped interface in the conical region facilitates stress relaxation by redistributing concentrated loads. This effect is illustrated in Panels (B) and (D) of Figure 7, where the stepped interface allows stress to be shared across multiple regions, avoiding a single high-stress concentration point.Increased contact area.The complex shape of the conical structure increases the contact surface area between the branch and the main stem, which aids in load dispersion and reduces localized stress.Fiber orientation and layered structure formation.
In the conical region, wood fibers from the branch and stem reorient, forming interwoven and interlocking structures. This interwoven fiber arrangement enhances tensile and shear strength at the connection zone.Shigo [2] observed that the conical structure arises from the intergrown connection of branch and stem, where branch-tail fibers turn and align with main stem fibers, forming a “cross-layered structure” and “interpenetrating structure” on the same plane. These features contribute significantly to the connection strength and the overall structural integrity [4].In model C2, the rigid reinforcement at the conical terminal enables smooth stress transmission without concentration. This adjustment prevents failure at the connection interface, which would otherwise occur.Gradual material transitions.The interwoven branch–stem structure, combined with the resinous or colloidal material in the darker boundary regions, forms a transitional layer between two relatively rigid structures. This material transition acts like a flexible cartilage layer between bones, reducing the risk of damage by smoothing stress distribution across the connection area and minimizing stress concentration at abrupt interfaces (Figure 7D).

**Figure 7 plants-14-00167-f007:**
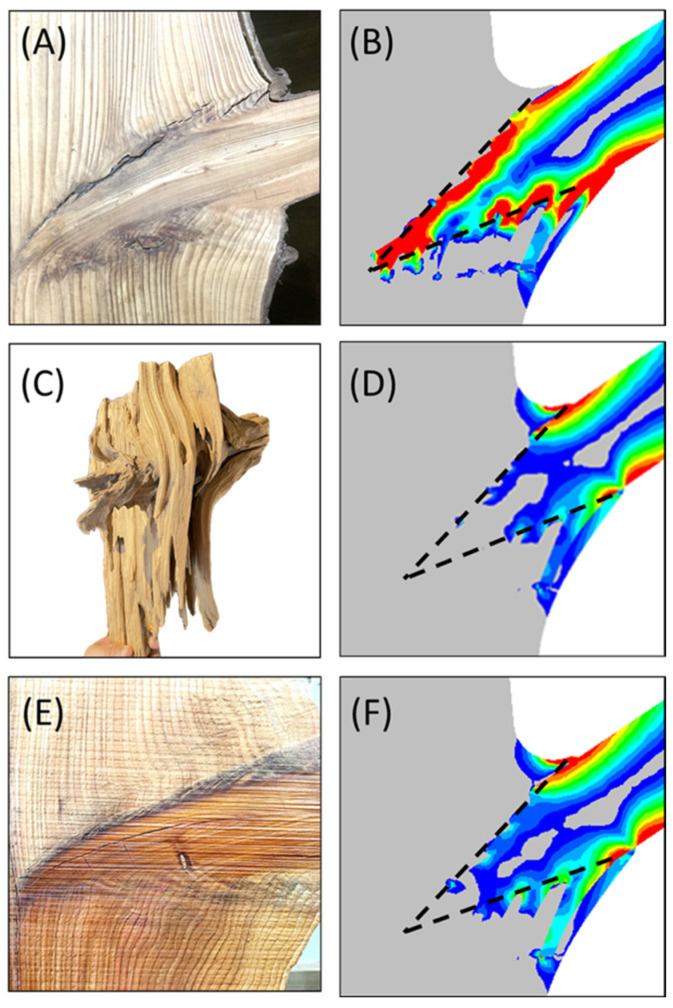
Panels (**A**,**B**) illustrate the stress concentration phenomenon and the resulting failure at the interface. Panels (**C**,**D**) depict the strengthening effects and the stress relaxation achieved through the complex interface structure. Panels (**E**,**F**) further demonstrate the enhanced support capacity of branches following resinous wood formation. Dashed lines indicate the boundary between the tapered conical structure and the main trunk structure. Red highlights high-stress regions, while blue and gray indicate areas of low stress.

#### 3.2.4. Mechanical Effects of Branch Heartwood Formation and Material Hardening

According to Garcia-Iruela et al. [9], resinous wood enhances material hardness. In model C3, which incorporates strengthened branch material properties, the displacement simulation under identical external loads demonstrated the smallest total displacement (Figure 6). This result confirms the mechanical benefits of resinous wood in improving branch support and structural rigidity.

### 3.3. Layered Response Strategies of Tree Fork Structures

The tree fork structure represents a two-part connection between branch and stem, formed by the alternating growth of fibers in different orientations (Figure 7C). In non-monolithic connections (B group), stress concentration often occurs at the interface, leading to structural vulnerability. To address this issue, trees employ a hierarchical strategy. First, shape optimization (Figure 2, A2) reduces surface and outer-layer damage. Additionally, the irregular conical shape, the formation of repair colloids, the redirection of branch-tail fibers, and the growth sequence collectively strengthen the connection interface, mitigating stress concentration (C group).

A comprehensive analysis of the mechanical strengthening strategies in tree forks reveals a synergistic effect involving external branch morphology, growth ring geometry, growth sequence, and cellular biochemical mechanisms. Trees employ the following strategies to enhance branch structural integrity:Optimizing external branch morphology.This improves surface-stress distribution and reduces stress concentration risks.Conical interface shape.The non-linear, stepped interface alleviates stress concentration (Figure 7B,D).Increased connection area.The conical shape enlarges the contact area between branch and stem, aiding in load distribution and reducing localized stress.Gradual stress transition.The conical structure enables smooth stress transmission across the connection zone, minimizing stress concentration from abrupt sectional changes.Fiber reorientation and layered structure formation.Alternating and interlocking fiber arrangements increase the tensile and the shear strength at the junction.Material strengthening and repair.Biochemical changes in cellular materials promote repair mechanisms and enhance material hardness.

These strategies collectively improve the mechanical performance of tree forks, reducing the risk of structural failure.

## 4. Materials and Methods

In this study, we employed a unique methodological design that incrementally added factors for discussion. Starting with external shape characteristics, we progressively introduced structural features, such as internal conical structures, to systematically evaluate their individual and combined effects on the mechanical stability of tree forks.

To minimize the interference of complex variables, such as fiber orientation, annual rings, and heterogeneous material properties, these influences were controlled by focusing on specific structural factors. By establishing controlled variable models, we directly assessed the contributions of these features to tree fork mechanics.

The following sections provide detailed descriptions of the materials, the model definitions, the preprocessing steps (including mesh generation and boundary condition settings), and the postprocessing procedures (such as stress and strain analysis and key data extraction). The overall experimental workflow is illustrated in Figure 8.

### 4.1. Sample Acquisition and Morphological Feature Extraction

To investigate the effects of the internal conical structure at branch–stem junctions on overall mechanical performance, Acer (maple) was selected as the morphological reference (Figure 9). As a common broadleaf species, Acer features distinct and easily observable branch–stem connections, providing clear geometric characteristics for analysis. While the external geometry of the models in this study was directly based on actual Acer specimens, the material properties used in subsequent numerical simulations were simplified to homogeneous isotropic parameters. As mentioned earlier, this simplification eliminates other variables, allowing us to focus on the direct effects of the conical structure and its morphological features on the mechanical behavior of tree forks.

For the acquisition of external geometric features, we used the Revopoint POP 2 3D scanner (Revopoint 3D Technologies Inc., Xi’an and Shenzhen, China; Figure 9, left image) to conduct high-precision laser scanning of the Acer samples. The scanning process generated accurate surface point cloud data, which served as a reliable geometric foundation for subsequent CAD modeling.

The internal structure was shaped based on direct observations and Shigo’s theoretical framework [1]. Notably, although Acer guided the external geometry, we did not use Acer’s internal structure (nor the same material parameters) as the standard for our model. Instead, a broader, simplified framework was employed to capture fundamental morphological features observable in various species. In dissected decayed-branch–stem specimens, a conical core structure was clearly identified (Figure 10a,b). This structure is not a single smooth surface but instead consists of multiple stepped truncated cones that decrease in size as they transition from the branch toward the main stem (Figure 10c). Beneath the conical core, horizontally extending branch fibers were observed to bend and reorient, eventually aligning with the vertical fibers of the main stem (Figure 10d).

Based on these observations, the models incorporated the stepped conical structure and branch-tail fiber features to more accurately simulate the natural mechanical behavior of tree forks.

### 4.2. Modeling of Branch–Stem Junctions

To systematically evaluate the effects of branch–stem junctions on the mechanical performance of tree forks, a series of 3D models were developed based on observed morphological features (Figure 11). These models include control and experimental groups to isolate the impact of specific structural factors.

#### 4.2.1. Model Classification and Structural Variations

Groups A and B primarily focus on differences in external geometric features, while Group C introduces refined internal structural variations based on the same baseline dimensions (Table 1). Although C2 and C3 share the same external geometry as C1, their internal boundary conditions and material properties differ. Further details on material settings and conditions are provided in Section 4.3.2 and Section 4.3.3.

#### 4.2.2. Model Dimensions and Structural Design

The model dimensions are categorized into external and internal components, described separately for cylindrical and fork-shaped configurations.

For the cylindrical models (A1, B1, B2), the dimensions were derived from the point cloud data: the main stem has a height of 300 mm, with the branch length measuring 240 mm and a connection angle of 55°. The junction occurs at 105 mm along the height of the main stem, while the branch tip diameter is 55 mm, and the main stem base diameter is 105 mm.

For the tree fork models (A2 and Group C), the external geometry of the junction was manually reconstructed based on the point cloud data, with a slight enlargement of the main stem base diameter to 130 mm. Within Group C, the internal features include a stepped conical structure that progressively contracts and incorporates fiber reorientation with distinct branch tail characteristics.

### 4.3. Preprocessing

The preprocessing phase involved establishing the applicability of FEA and configuring essential parameters, including material properties, boundary conditions, and mesh generation.

#### 4.3.1. Applicability of FEA

Given the complexity and heterogeneity of natural trees, isolating causal relationships between structural factors under real-world conditions is challenging. Computer simulations allow us to control variables and systematically study the mechanical behavior at branch–stem junctions in a well-defined and reproducible environment.

Finite element analysis (FEA) has been widely applied in tree structure research to evaluate stress distribution and mechanical performance, demonstrating its suitability for modeling complex biological systems [1,3,4,15,16,17,18]. FEA enables the discretization of irregular surfaces and complex structures, dividing the tree fork geometry into smaller elements to analyze stress and deformation. By using systematically designed models that differ by a single factor (as detailed in Section 4.2.1), this approach overcomes experimental challenges posed by sample variability and provides theoretical insights into the functional mechanisms of the conical structure.

#### 4.3.2. Material Properties

Both the experimental and control models were assigned isotropic material parameters to eliminate the effects of fiber orientation, annual ring distribution, and other material heterogeneities. This simplification allows the analysis to focus on the influence of tree fork structural features—particularly the conical structure—on mechanical behavior. Under this uniform material framework, the independent effects of structural and material factors can be systematically separated and observed, ensuring that the observed mechanical differences arise solely from the predefined variables rather than uncontrolled complexities.

Material parameters were primarily obtained from the SOLIDWORKS (Version 2023) built-in material library, which references authoritative sources such as the ASM International Metals Handbook Desk Edition (2nd Edition). For this study, the material properties were defined as follows: elastic modulus (E): 9 GPa, Poisson’s ratio (ν): 0.4, and density (ρ): 410 kg/m^3^. The shear modulus (G) was derived through calculations rather than directly provided by the material library.

For all dual-structure models, the above parameters were applied. For C2, a rigid connection was applied at the interface of the conical structure to the main trunk. In contrast, for C3, the elastic modulus was increased to 18 GPa to simulate the effect of material hardening, such as resinification or heartwood formation [8].

#### 4.3.3. Mesh

High-quality meshes were used for FEA. The Jacobian point number was set to 16 to ensure mesh quality. Element aspect ratios were maintained below 1:10 to ensure the convergence of the results. To validate the numerical stability, the mesh density was carefully adjusted and tested, confirming that all models produced consistent and convergent results under the applied boundary conditions. The numbers of nodes and elements in each model are presented in Table 2.

### 4.4. Postprocessing and Data Analysis

In this study, we used SOLIDWORKS Simulation 2020 (Dassault Systèmes SolidWorks Corporation, Waltham, MA, USA) to perform the stress analyses. The results were compared using visual representations and data curves.

#### 4.4.1. Stress and Displacement Diagrams

Figure 2 presents the stress diagrams, showing data distributions for both the surface and the lower surface, highlighting variations in equivalent stress and shear stress. The curve corresponding to the internal region depicts stress, shear stress, and total displacement along the symmetry plane cross-section of the structure.

#### 4.4.2. Numerical Result Profiles

To compare mechanical performance between the structural models, we extracted data from key regions and generated stress–position curves (Figure 3). Because of the structural symmetry and boundary conditions, considerable changes in stress and displacement were concentrated along the symmetry plane; thus, our analysis focused on the stress distribution in this region.

To ensure comparability of the results, we uniformly processed data from the surface, the lower surface, and the key internal region (Figure 12). The data extraction path began at the branch–stem junction and extended along the direction of the branch. To eliminate random errors in the data, we used the Lowess method for locally weighted regression, enhancing the smoothness and readability of the curves.

The data were projected onto the x-axis, and three key positions were identified as follows: x = 0 mm (starting point of the conical structure), x = 42 mm (branch–stem junction), and x = 78 mm (branch collar position). These positions were instrumental in understanding variations in stress and displacement with respect to position. However, the focus of this study was on overall structure; thus, we did not delve deeply into each x value, rather we included them as reference points for comparing the structural models.

### 4.5. Experimental Limitations

This study established models based on the morphological features of Acer tree forks; however, these features do not necessarily represent the tree fork morphology of all species or under varying environmental conditions. Therefore, caution is advised when generalizing the results to other contexts.

Additionally, the material parameters used in this study were obtained from built-in software libraries to ensure consistency in comparison conditions. However, this approach may not accurately reflect the anisotropic and heterogeneous nature of real wood. By assuming wood as an isotropic material, we intentionally excluded the effects of fiber orientation and anatomical variability, which inherently limits the ability to fully capture the complex mechanical behavior of wood.

Moreover, the study employed a linear elastic assumption, which does not account for the material yielding or for failure behavior under extreme loading. In practical applications, when stress exceeds a certain threshold, wood fibers may undergo irreversible deformation or fracture. Furthermore, external environmental factors, such as wind, rainfall, and snow accumulation, were not considered in the current analysis.

Future research should incorporate more accurate wood material properties, including anisotropy and failure criteria, and expand the scope of environmental loads. Such refinements will allow for a more comprehensive and realistic understanding of the mechanical behavior of tree forks across diverse tree species and conditions.

## 5. Conclusions

This study explored the effects of the external shape and internal structure of tree forks on their mechanical performance, elucidating their critical functions and reinforcement strategies within the tree’s structural framework. Based on experimental analyses and numerical simulations, the following conclusions were reached:Influence of fork shape on stress distribution.

The external shape of a tree fork greatly affects the distribution of surface stress. Models with identical shapes exhibited similar stress patterns, whereas those incorporating tree fork features displayed distinct stress distributions. Such shape characteristics effectively disperse load, reducing the risk of stress concentration and subsequent structural failure within the trunk.

2.Stress conduction and concentration in the conical structure.

The conical structure formed at the trunk–branch junction guides stress transfer but also creates localized stress concentrations at the interface. Comparisons between plate-like and conical-insertion models showed that the latter generated more pronounced internal stress concentrations, consistent with observed damage patterns in actual trees.

3.Stress mitigation effects of the conical structure.

The conical structure offers multiple mechanical advantages, including a nonlinear, stepwise interface, enlarged contact areas, altered fiber orientations, and stratified configurations. These features collectively promote load dispersion, mitigate stress concentrations, and enhance both tensile and shear strength. Interface-enhanced models confirmed that the conical structure effectively diminishes stress accumulation, thereby protecting the tree against structural failure.

4.Influence of material properties.

Resinification and heartwood formation within tree forks increase material hardness and the load-bearing capacity. Simulations involving resinous wood in branches indicated reduced displacement and improved structural stability. These findings suggest that material strengthening mechanisms naturally employed by trees greatly improve the mechanical performance and longevity of tree forks.

5.Multilevel strategies in tree fork optimization.

Trees integrate a suite of multilevel strategies to optimize fork structures, including external shape optimization, targeted stress mitigation through conical formations, adjustments in fiber orientation, and strategic material reinforcement. These approaches operate synergistically to enhance overall mechanical performance and reduce the likelihood of junction failure.

In summary, this study provides comprehensive insights into the integrated reinforcement mechanisms of tree forks, including shape design, internal structural features, and interface properties, thereby offering a solid scientific foundation for evaluating and maintaining tree structural integrity. While the current research has focused on conical characteristics, future investigations should delve more deeply into the anisotropic, heterogeneous, and multilayered nature of tree materials to further illuminate how trees adapt and maintain mechanical resilience under varying natural conditions.

## Figures and Tables

**Figure 1 plants-14-00167-f001:**
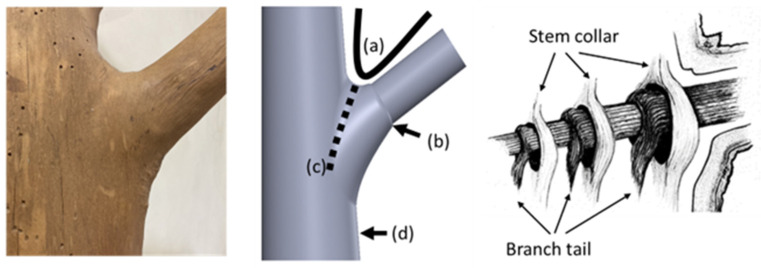
Structural analysis of a branch–stem junction. The (**left**) image shows the external morphology of a beech tree’s branch–trunk junction, highlighting the natural configuration at the tree fork. The (**middle**) image, adapted from a study conducted by Liu et al. [4], indicates four key features of the junction that contribute to structural stability: (a) the U-shaped connection at the tree fork, (b) a collar around the base of the branch, (c) a bark ridge along the junction, and (d) a thickened area below the trunk. The (**right**) image, based on a model developed by Shigo [2], shows the intertwined internal structure between the branch collar and the trunk collar (indicated by arrows), forming a unique interlocking fiber system that enhances the mechanical stability of the branch–stem junction.

**Figure 3 plants-14-00167-f003:**
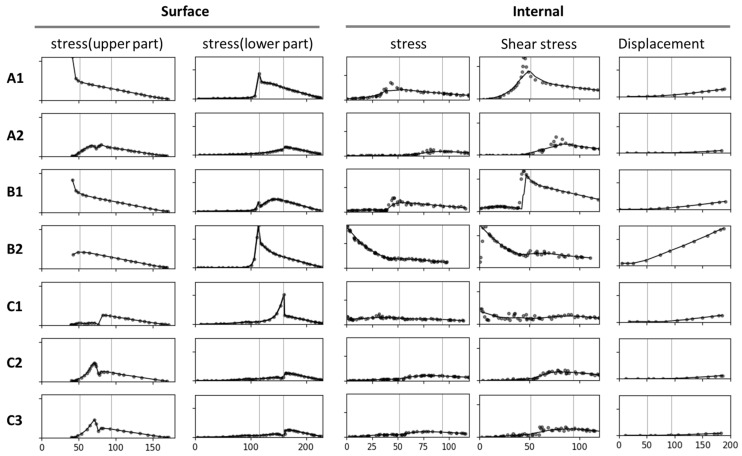
Stress–position curves for key areas analyzed using seven models. The figure indicates changes in stress, shear stress, and total displacement across the surface, the lower surface, and the internal region. Curves in the same column have the same y-axis and x-axis; no specific ranges or units are indicated. The curves indicate the stress and displacement at the branch–stem junction for various structural designs.

**Figure 4 plants-14-00167-f004:**
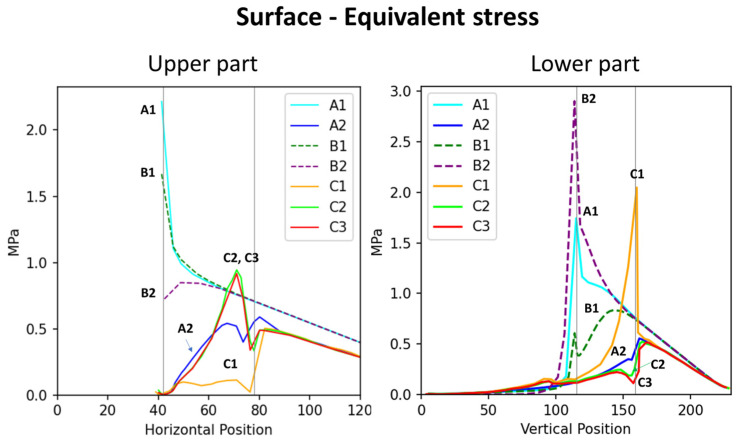
Distribution of stress at the surface and lower surface regions of various branch–stem junctions. The (**left**) graph depicts the distribution of surface stress across horizontal positions, whereas the (**right**) graph depicts the distribution of lower surface stress across vertical positions. Vertical lines indicate two crucial points: the right line indicates the branch collar boundary, whereas the left line indicates the branch–stem junction. The length of the x-axis differs between the graphs because of differences in the curve extraction path.

**Figure 5 plants-14-00167-f005:**
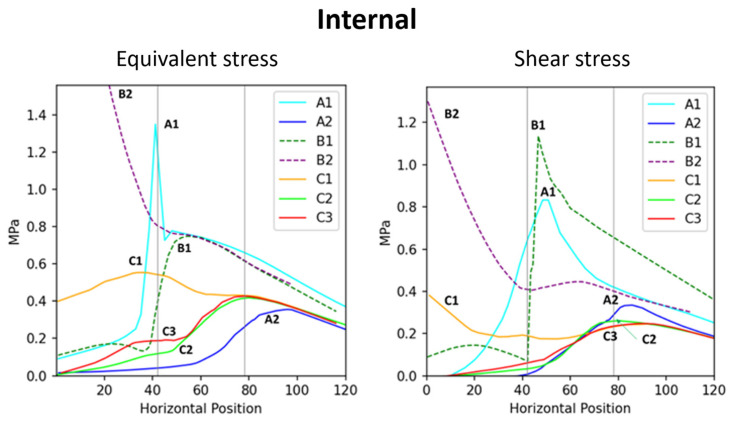
Correlation between the internal stress and the position across branch–stem junctions. The (**left**) diagram indicates internal stress, whereas the (**right**) diagram indicates internal shear stress.

**Figure 6 plants-14-00167-f006:**
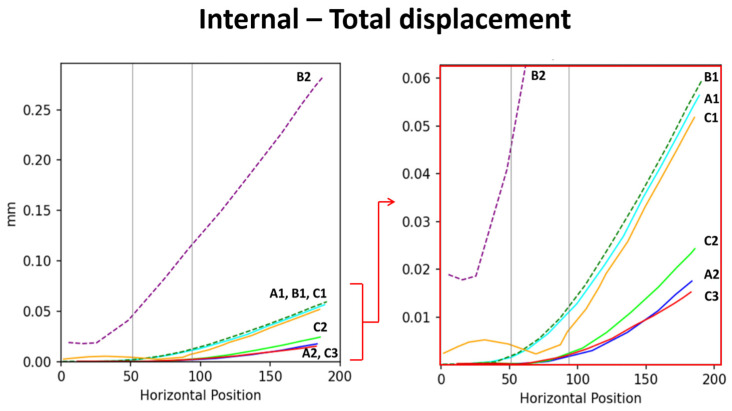
Displacement scenarios of each model. The (**left**) image shows the overall displacement distribution, with the red arrow indicating the localized region that is magnified in the (**right**) image presents localized displacement results. Based on the displacement data, the support capacity of the models is ranked from highest to lowest as follows: B2 > B1 > A1 > C1 >> C2 > A2 > C3.

**Figure 8 plants-14-00167-f008:**
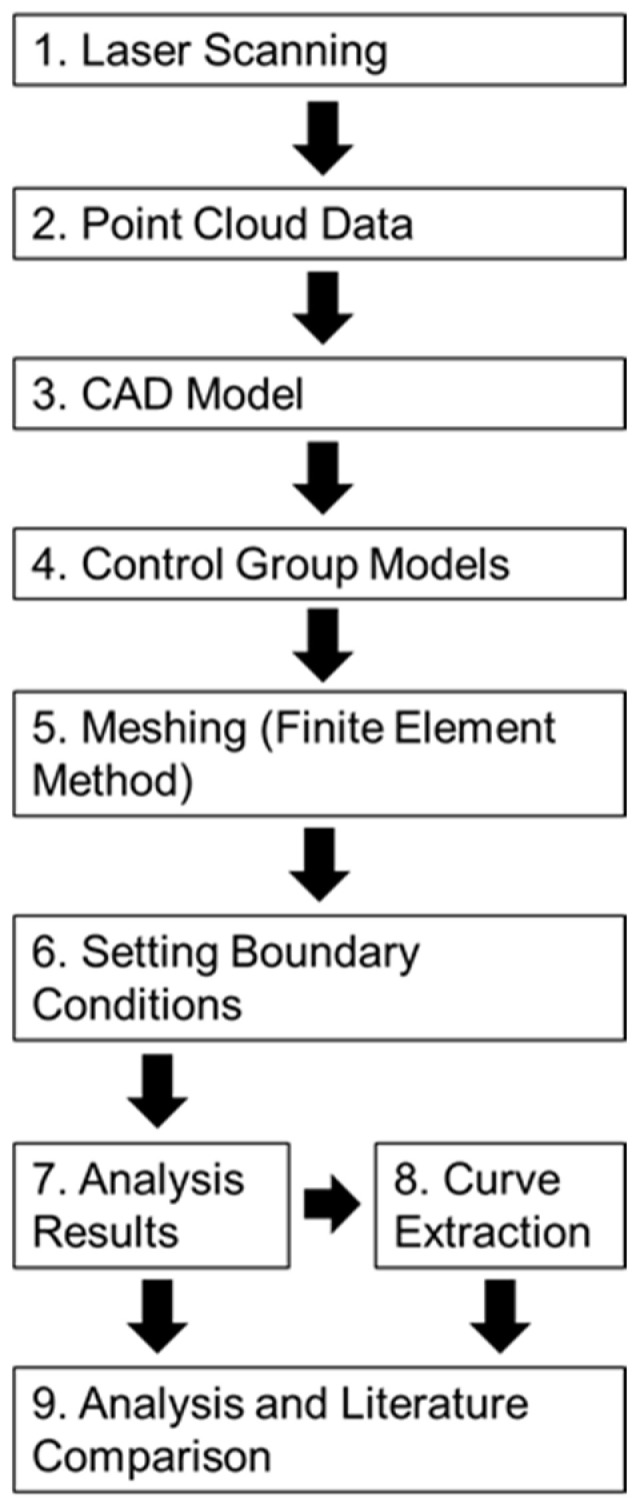
Flowchart depicting the experimental process.

**Figure 9 plants-14-00167-f009:**
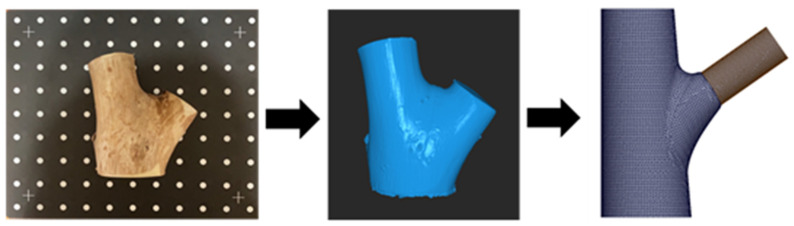
Experimental materials and the modeling process for branch–stem junctions. The (**left**) image shows the original wood sample, scanned using a Revopoint POP 2 scanner (Revopoint 3D Technologies Inc., Xi’an and Shenzhen, China). The (**middle**) image shows the point cloud data model generated using data from the scan. The (**right**) image shows the branch–stem junction structure model developed for finite element analysis based on the point cloud data.

**Figure 10 plants-14-00167-f010:**
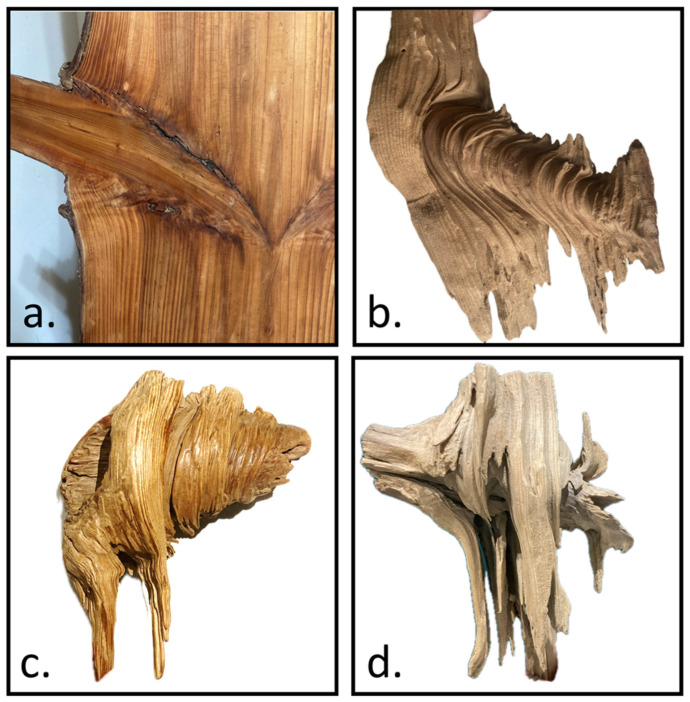
Illustrates the key morphological features of the branch–stem internal structure. (**a**) The original tissue configuration at the branch–stem junction. (**b**) The conical core structure revealed after decay of the surrounding material. (**c**) Multiple stepped truncated cones form the conical structure, decreasing in size as they transition from the branch toward the main stem. (**d**) Beneath the conical core, the branch fibers bend and reorient, eventually aligning with the main stem’s vertical fiber orientation.

**Figure 11 plants-14-00167-f011:**
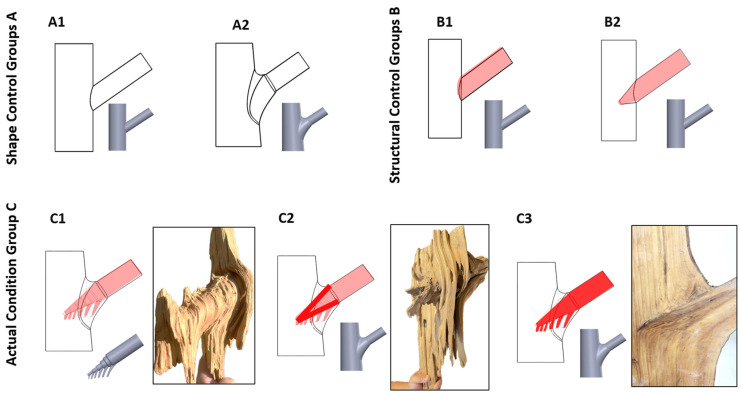
Branch–stem model design. The figure shows different branch–stem junction models. (**A1**) represents a cylindrical connection model without tree fork characteristics, simulating a basic branch–stem junction. (**A2**) incorporates a tree fork connection. (**B1**) simulates a double structure with surface attachment. (**B2**) represents a tapered insertion model simulating the independent connection between the trunk and the branch. (**C1**) mimics the natural external shape and tapered insertion of a tree but does not include internal connections formed through alternating growth. (**C2**) has the same external shape as C1 but simulates the internal reinforcement resulting from alternating growth. (**C3**) accounts for the material hardening properties of resinification and heartwood formation in the side branch, simulating the effect of structural strengthening.

**Figure 12 plants-14-00167-f012:**
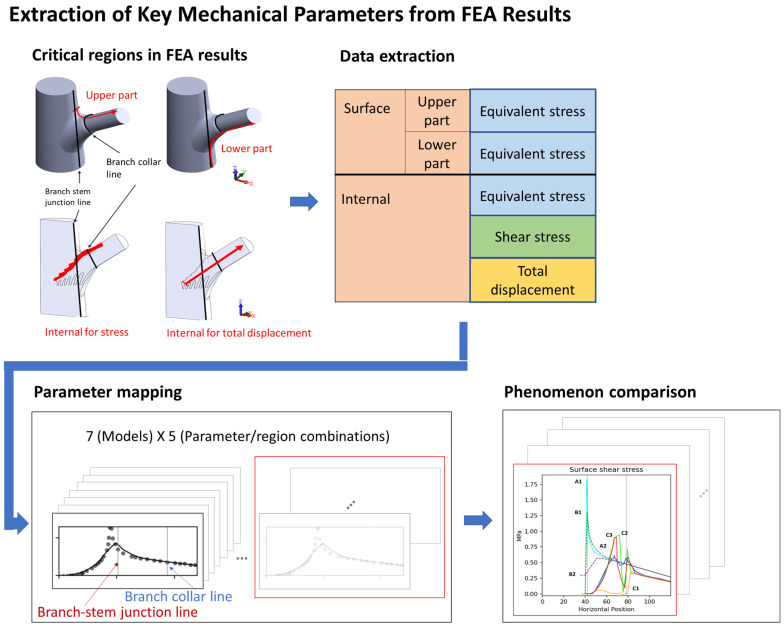
This figure illustrates the process of extracting key mechanical parameters from finite element analysis (FEA) results, highlighting critical sampling regions and data mapping. Red-highlighted areas denote specific structural locations where stress and displacement values were recorded, underscoring their importance in evaluating mechanical performance. Sampling was conducted at four distinct locations: two on the surface—the upper surface (a) and lower surface (b)—and two internal regions—the interface area (c) and the branch axis (d). The extracted data were subsequently mapped and visualized to compare stress patterns and total displacement across models.

**Table 1 plants-14-00167-t001:** Model designs and experimental groups.

Group	Model Name	Structural Design	Tree Fork Characteristics	Material Properties
Group A	A1	Cylindrical connection	None	Isotropic material
	A2	Tree fork connection	Tree fork shape	
Group B	B1	Surface attachment	None	
	B2	Tapered insertion		
Group C	C1	Conical model	Tree fork shape	
	C2	Interface connection enhancement		Considering material hardening (resinification and heartwood formation)
	C3	Hardened model		

**Table 2 plants-14-00167-t002:** Numbers of nodes and elements in each model group.

Group	Number of Nodes	Number of Elements
A1	303,985	213,086
A2	452,581	405,068
B1	185,278	128,046
B2	332,524	233,707
C1	452,581	405,068
C2	452,581	405,068
C3	452,581	405,068

## Data Availability

The original contributions presented in this study are included in the article. Further inquiries can be directed to the corresponding author.

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
