# Peer review of "Mechanical Properties and Optimization Strategies of Tree Fork Structures"

_plants, 2025, doi:10.3390/plants14020167_

Round 1
Reviewer 1 Report
Comments and Suggestions for Authors
Topic of the manuscript is interesting. The aims are clearly stated at the end of introduction.
Some problems arose during reading of the manuscript. The chapters of 2. Results, 3. Discussion, 4. Materials and methods are not in usual arrangement.
The properties used in calculations indicate the composition of “fork” is isotropic material rather than Acer wood. Later on, authors admit the isotropic character of acer wood, line 375. From the material point of view, density value is too low as well as shear modulus. Moreover, modulus of elasticity, shear modulus and Poisson ratio do not coincide for isotropic material together.
The force is defined at line 375, but the dimensions of “fork” do not.
The magnitude of 2, 3, 7, 10 figures are low. The figures are not readable.
Author Response
Reply to Reviewers of their Comments and Suggestions from Authors
We had reviewed all comments and responded all suggestions to reply as follows one by one. The following revisions in the manuscript are marked in red color.
|
Main concerns: |
(I) Ensure all references are relevant to the content of the manuscript. (II) Highlight any revisions to the manuscript, so editors and reviewers can see any changes made. (III) Provide a cover letter to respond to the reviewers’ comments and explain, point by point, the details of the manuscript revisions. (IV) If the reviewer(s) recommended references, critically analyze them to ensure that their inclusion would enhance your manuscript. If you believe these references are unnecessary, you should not include them. (V) If you found it impossible to address certain comments in the review reports, include an explanation in your appeal.. |
|
Reply |
1. We have thoroughly reviewed all the references cited in the manuscript to ensure their direct relevance to our study’s objectives and findings. 2. Highlighting Revisions: All changes made in the revised manuscript were highlighted for easy identification by the editors and reviewers (red color). 3. Cover Letter: We provided a cover letter that explains, point by point, the revisions made to the manuscript and responses to the referees' comments. 4. We meticulously responded to each of the reviewer's comments and accordingly revised the manuscript. 5. Review Process: The revised version of the manuscript, along with the cover letter and highlighted changes, was submitted for review by the editors and reviewers |
Reviewer #1:
Comments and Suggestions for Authors
|
Main concerns: |
1. Some problems arose during reading of the manuscript. The chapters of 2. Results, 3. Discussion, 4. Materials and methods are not in usual arrangement. |
|
Reply |
Thank you for your comment. The chapter arrangement follows the journal's specific template: Introduction, Results, Discussion, Methods, and Conclusion. |
|
Main concerns: |
2. The properties used in calculations indicate the composition of “fork” is isotropic material rather than Acer wood. Later on, authors admit the isotropic character of acer wood, line 375. |
|
Reply |
This study reconstructed the external shape using point cloud data from Acer (see Section 4.1) while adopting a homogeneous material assumption (see Section 4.3.2). This approach was chosen to eliminate the interference of material heterogeneity (e.g., fiber orientation) and focus on the mechanical behavior influenced by structural features. The goal of this study is to understand the mechanical mechanisms of geometry and structure rather than fully simulate real wood. We have acknowledged this limitation in Section 4.6, and the use of non-homogeneous materials remains a topic for future discussion. Thank you for your feedback. |
|
Revised lines code |
1) Section 4.1 2) Section 4.3.2 (page 1, line 3-4). 3) Section 4.6 |
|
Main concerns: |
3. From the material point of view, density value is too low as well as shear modulus. Moreover, modulus of elasticity, shear modulus and Poisson ratio do not coincide for isotropic material together. |
|
Reply |
Thank you for highlighting the issues regarding material parameters. Upon review, the original shear modulus was corrected to address a calculation error, and the updated value now accurately reflects the isotropic assumption.
Regarding the lower density and shear modulus values, these were sourced from the SOLIDWORKS material library. To ensure focus on structural features, we utilized a methodology of single-factor difference models, systematically increasing factors for comparison. This approach minimizes the influence of material properties and other non-essential variables, maintaining clarity in evaluating mechanical effects. |
|
Revised lines code |
4) 4.2.1 Model classification and structural variations 5) 4.3.2 Material Properties 6) 4.6 Experimental Limitations |
|
Main concerns: |
4. The force is defined at line 375, but the dimensions of “fork” do not. |
|
Reply |
Thank you for your comment. The dimensions of the “fork” are defined in Section 4.2.2 (Model dimensions and structural design), where we provide detailed descriptions of the main trunk and branch sizes, including height, length, diameters, and connection angles. |
|
Revised lines code |
7) 4.2.2 Model dimensions and structural design (Line327-339) |
|
Main concerns: |
5. The magnitude of 2, 3, 7, 10 figures are low. The figures are not readable. |
|
Reply |
We apologize for the inconvenience caused. The figures (2, 3, 7, 10) have been revised and replaced with higher-resolution versions, ensuring each exceeds 1000 pixels for better clarity and readability. Additionally, we have improved the presentation of several figures to enhance their visual clarity and ease of interpretation. |
|
Revised lines code |
8) Figure 2, 3, 7, 10, 11, 12 |
Reviewer 2 Report
Comments and Suggestions for Authors
Effects of Shape, Structure, and Material on the Mechanical Properties of Tree Forks: A Three-Level Framework for Structural Stability Assessment
Article is very interesting, but it requires to provide some corrections. The weak point is no real validation of the model, no certainty about its correctness. Additionally, significant simplification - assumption regarding homogeneity of material critically reduces scientific soundness of the article. conclusions are repeated many times in the work, and should be included in the summary of the work. Additionally, in the reviewer's opinion - conclusions drawn from the simulation are not entirely conclusions, but only a description of natural mechanisms related to plant biology. The conclusions should be corrected.
Line 36 – were the simulations validated ? what was the methodology of validation?
Line 80, 209 – why is the typical chapter order reversed? the logical system is materials and methods and only then results
Line 88,89 etc. – in case of grayscale print it is not possible to analyze figures
Lines 88-93 and 97-105 – there is no need to repeat the same information twice
Line 94 – what exactly are surface conditions and internal conditions?
Line 87 – what is the justification for using Mises stresses, which are reduced stresses and are used for homogeneous materials? wood - as is commonly known - is not a homogeneous material and the stress components should be analyzed in individual anatomical directions, or at least in two directions - longitudinal and tangential
Line 130 - what characterizes the improved distribution of stress
Line 161 - How is resinification and heartwood formation reflected in the model?
Line 201- fig is too small
Lines 178-200 – it is not the right place for a summary and conclusion
Line 209 – as previously written – should be before results
Lines 223 – 245 - does not contribute much to the topic, unnecessary introduction to numerical simulations
Line 259 - What was involved in simplifying the model?
Line 278, fig 10- too small to analyse
Line 288 – table 1 and material properties – the basic weakness of the authors' considerations is the assumption of material homogeneity ! the assumption of homogeneity leads to erroneous results, supplemented by the analysis of Misses stresses is a methodological error
Line 310 – what material database ? literature ? test results?
Line 315 - What is the dounar method?
Line 318 - How is the interpenetration of elements designed? What are the boundary conditions for a log and branch connection?
Line 376 – please explain – “it used plastic materials for simulations”
Line 398- further reinforcement of this structure made by whom?
Line 410 – in what specific way? Please specify
Line 432 – References – very limited, many pieces of literature from the 1900s
Author Response
Reply to Reviewers of their Comments and Suggestions from Authors
We had reviewed all comments and responded all suggestions to reply as follows one by one. The following revisions in the manuscript are marked in red color.
|
Main concerns: |
(I) Ensure all references are relevant to the content of the manuscript. (II) Highlight any revisions to the manuscript, so editors and reviewers can see any changes made. (III) Provide a cover letter to respond to the reviewers’ comments and explain, point by point, the details of the manuscript revisions. (IV) If the reviewer(s) recommended references, critically analyze them to ensure that their inclusion would enhance your manuscript. If you believe these references are unnecessary, you should not include them. (V) If you found it impossible to address certain comments in the review reports, include an explanation in your appeal.. |
|
Reply |
1. We have thoroughly reviewed all the references cited in the manuscript to ensure their direct relevance to our study’s objectives and findings. 2. Highlighting Revisions: All changes made in the revised manuscript were highlighted for easy identification by the editors and reviewers (red color). 3. Cover Letter: We provided a cover letter that explains, point by point, the revisions made to the manuscript and responses to the referees' comments. 4. We meticulously responded to each of the reviewer's comments and accordingly revised the manuscript. 5. Review Process: The revised version of the manuscript, along with the cover letter and highlighted changes, was submitted for review by the editors and reviewers |
Reviewer #2:
Comments and Suggestions for Authors
|
Main concerns: |
1. The weak point is no real validation of the model, no certainty about its correctness. 2. Line 36 – were the simulations validated ? what was the methodology of validation? |
|
Reply |
Thank you for your question. We validated the accuracy of the results through the following approaches: 1. Morphology and Structure Source: The external morphology and internal structure of the models were based on tree specimen observations (see Figure 10) and classical theories such as the Shigo model. 2. Research Strategy: A single-factor control strategy was employed, where only specific structural features were modified while all other conditions remained consistent, ensuring scientific rigor and reliability in model comparisons. 3. Result Comparison: The stress and displacement differences across models were analyzed, and the simulation results aligned with the mechanical failure locations reported in existing literature (see references 3, 8, and 14). 4. Numerical Validation: Finite element analysis was conducted with strict control of mesh density to ensure the convergence and stability of the results. |
|
Revised lines code |
9) Line289-298 10) Line305 (section 4.2. Modeling of Branch–stem Junctions) 11) Line 379 (4.3.4 Mesh) |
|
Main concerns: |
3. Additionally, significant simplification - assumption regarding homogeneity of material critically reduces scientific soundness of the article. |
|
Reply |
Thank you for your valuable feedback. This study does not aim to simplify the complexity of trees but focuses on the feature extraction and mechanical analysis of conical structures. Tree structures encompass numerous complex factors, such as material heterogeneity, fiber orientation, and biological characteristics.
Our decision to concentrate on the geometric and mechanical performance of conical structures stems from the core objectives of this study. This does not diminish the importance of other factors, which will be explored in greater depth in future research.
We appreciate your highlighting this critical point. We have revised the manuscript to ensure the research objectives are clearly stated, emphasizing that this study provides a foundational analysis for tree structural mechanics and supports the expansion of future investigations. |
|
Revised lines code |
12) Line 2(Topic) 13) Section 4.3.2 14) Section 4.6 |
|
Main concerns: |
4. Conclusions are repeated many times in the work, and should be included in the summary of the work. |
|
Reply |
We have reorganized and significantly revised the presentation. Thank you for your valuable suggestions. |
|
Revised lines code |
15) Line 151 (Section 3. Discussion) 16) Line 444 (Section 5. Conclusions) |
|
Main concerns: |
5. In the reviewer's opinion - conclusions drawn from the simulation are not entirely conclusions, but only a description of natural mechanisms related to plant biology. The conclusions should be corrected.
|
|
Reply |
Thank you for your feedback. Based on your suggestion, we have made extensive revisions to the conclusion and discussion sections. Additionally, we have refined the abstract and main topics to ensure consistency throughout the manuscript.. |
|
Revised lines code |
17) Title (page 1, line 3 through line 4). 18) Line 141-205 (Page 6-7, Section 3. Discussion) 19) Line437-478 (Page 15-16, Section 5. Conclusions) |
|
Main concerns: |
6. Line 80, 209 – why is the typical chapter order reversed? the logical system is materials and methods and only then results 7. Lines 178-200 – it is not the right place for a summary and conclusion 8. Line 209 – as previously written – should be before results |
|
Reply |
Thank you for your comment; the chapter arrangement follows the journal’s template guidelines. |
|
Main concerns: |
9. Line 88,89 etc. – in case of grayscale print it is not possible to analyze figures |
|
Reply |
Thank you for your comment. This journal is published electronically, allowing readers to view the figures in full color. Additionally, we have used distinct line styles and markers in the legends to differentiate data, ensuring readability even under grayscale printing conditions. |
|
Main concerns: |
10. Lines 88-93 and 97-105 – there is no need to repeat the same information twice |
|
Reply |
Thank you for your feedback. We have revised and removed the redundant content. |
|
Revised lines code |
20) Line91-94 21) Line96-99 |
|
Main concerns: |
11. Line 94 – what exactly are surface conditions and internal conditions? |
|
Reply |
Thank you, the explanation has been supplemented. |
|
Revised lines code |
22) Line91-98 |
|
Main concerns: |
12. Line 87 – what is the justification for using Mises stresses, which are reduced stresses and are used for homogeneous materials? wood - as is commonly known - is not a homogeneous material and the stress components should be analyzed in individual anatomical directions, or at least in two directions - longitudinal and tangential |
|
Reply |
In terms of stress representation, external tensile and compressive stresses primarily occur on the surface, which is why equivalent stress (Mises stress) was used. Shear stress typically arises within the internal structure, so internal shear stress and total displacement observations were included in the model to comprehensively illustrate the mechanical behavior of the structure both externally and internally.
The manuscript has been revised to clarify this rationale and the study’s focus. |
|
Revised lines code |
23) Section 2.1 Results of Stress Analyses 24) Section 4.3.2 25) Section 4.6 (Experimental Limitations) |
|
Main concerns: |
13. Line 130 - what characterizes the improved distribution of stress |
|
Reply |
Thank you for your question.We have significantly revised the discussion and conclusion sections to address the characteristics of the improved stress distribution in detail. Thank you for your valuable feedback. The improved stress distribution is characterized by: 1. Reduced Stress Concentration: Stress is minimized at critical points, lowering the risk of structural failure. 2. Guided Stress Flow: Stress is directed to specific regions that can better handle the load. 3. Load Dispersion: A larger contact area, particularly within the conical structures of the tree fork, helps distribute the load more evenly. 4. Interlocking Fiber Arrangements: The interwoven fibers enhance connection strength and stability. 5. Gradual Geometric Transitions: Smooth transitions in shape and material properties reduce abrupt stress changes, improving mechanical performance. These points are elaborated in the discussion and conclusion sections. |
|
Revised lines code |
26) Line 141-205 (Page 6-7, Section 3. Discussion) 27) Line 433-474 (Page 15-16, Section 5. Conclusions) |
|
Main concerns: |
14. Line 161 - How is resinification and heartwood formation reflected in the model? |
|
Reply |
Thank you for your question. In Model C3, resinification and heartwood formation were simulated by increasing the elastic modulus of the side branch material to 18 GPa, based on sample observations and literature. This is described in Section 4.3.2 Material Properties and Section 4.2.1 Model classification and structural variations. |
|
Revised lines code |
28) Line307 (Page 11, Section 4.2.1) 29) Line369-372 (Page12, Section 4.3.2) |
|
Main concerns: |
15. Line 201- fig is too small 16. Line 278, fig 10- too small to analyse |
|
Reply |
We apologize for the inconvenience caused. The figures have been revised and replaced with higher-resolution versions, ensuring each exceeds 1000 pixels for better clarity and readability.
|
|
Revised lines code |
30) Figure 2, 3, 7, 10 |
|
Main concerns: |
17. Lines 223 – 245 - does not contribute much to the topic, unnecessary introduction to numerical simulations |
|
Reply |
Thank you for your suggestion. We have revised the section to focus on the application and advantages of FEA in studying branch–stem junctions, ensuring the content aligns more closely with the research topic. |
|
Revised lines code |
31) Line 341-353 (Page 12, Section 4.3.1 Applicability of FEA) |
|
Main concerns: |
18. Line 259 - What was involved in simplifying the model? |
|
Reply |
This study did not merely simplify the model but instead incrementally added factors for analysis. We first simulated the external shape characteristics, followed by the internal structures (e.g., conical features) to systematically assess their impact on mechanical behavior. |
|
Revised lines code |
32) Line 250-261 (Page9, Section 4. Materials and Methods) 33) Line 302-312 (Page11, Section 4.2. Modeling of Branch–stem Junctions) |
|
Main concerns: |
19. Line 288 – table 1 and material properties – the basic weakness of the authors' considerations is the assumption of material homogeneity ! the assumption of homogeneity leads to erroneous results, supplemented by the analysis of Misses stresses is a methodological error |
|
Reply |
The anisotropic fibers, layered structures, moisture variations, and heterogeneity of tree materials, along with species differences, make trees exceptionally challenging to simulate. However, our research method incrementally adds factors for discussion. We first examine the influence of external shape, then introduce structural factors, using comparative models to analyze these variations. Other tree-specific factors were not addressed in this study but are planned for the next phase of our research. Thank you for your insightful question. |
|
Revised lines code |
34) Line250-261(Page9, Section 4. Materials and Methods) |
|
Main concerns: |
20. Line 310 – what material database ? literature ? test results? |
|
Reply |
Thank you for your question. The material parameters used in this study were primarily sourced from the built-in material database of SOLIDWORKS, referencing authoritative sources such as the ASM International Metals Handbook, to provide a standardized material baseline. This approach allowed us to control variables and focus on the influence of structural features on mechanical performance. Additionally, in the C2 and C3 models, material adjustments were made based on insights from literature. For example, in the C3 model, we simulated the hardening properties of resinification and heartwood formation, increasing the elastic modulus to 18 GPa, as suggested by studies like Garcia-Iruela et al. These clarifications have been added to the revised manuscript for greater transparency and completeness. |
|
Revised lines code |
35) Line 211 (Page 7, Section 3.2.4 Mechanical effects of branch heartwood formation and material hardening) 36) Line358-376 (Page 12-13, Section 4.3.2 Material Properties) 37) Line367-372 (Page13, Section 4.3.2. Material Properties) |
|
Main concerns: |
21. Line 315 - What is the dounar method?Dounar |
|
Reply |
Thank you for your question. The mention of the Dounar method has been removed in the revised manuscript to avoid potential misunderstanding. The study focuses on the effects of structural and geometric differences on mechanical behavior, aligning with our research objectives. |
|
Main concerns: |
22. Line 318 - How is the interpenetration of elements designed? What are the boundary conditions for a log and branch connection? |
|
Reply |
Thank you for your question. The design of element interpenetration was based on observations and image records of actual specimens. The samples revealed that the downward curving branch tongue overlaps with the annual rings of the main trunk, forming a similar structure. This feature was replicated in the model to simulate interwoven behavior.
In the finite element analysis, the boundary conditions for the tree model were set as non-penetrating solid boundaries, ensuring that no numerical penetration or unrealistic deformation occurs during the simulation. |
|
Revised lines code |
38) Line 373-376 (Page 13, Section 4.3.2. Material Properties) |
|
Main concerns: |
23. Line 376 – please explain – “it used plastic materials for simulations” |
|
Reply |
Thank you for pointing this out. We have revised the text to consistently use "isotropic materials" |
|
Main concerns: |
24. Line 398- further reinforcement of this structure made by whom? |
|
Reply |
The boundary condition based on both observations and literature establishes a standardized method to assess how material hardening strengthens tree fork mechanics. This research integrates findings from Garcia-Iruela et al. [8], highlighting how resinification and heartwood development enhance hardness and load-bearing capacity. In the C3 model, the elastic modulus of the side branch was raised to 18 GPa to simulate these effects. |
|
Revised lines code |
39) Line369-372 (Page 13, Section 4.3.2. Material Properties) |
|
Main concerns: |
25. Line 410 – in what specific way? Please specify |
|
Reply |
Thank you for your question. The conical structure is a common feature in normal branch connections, but its internal stress distribution mechanisms were previously unclear. This study analyzed how the external shape and internal geometry of the fork structure respond to external loads through finite element simulations. The results revealed that the nonlinear, stepped interfaces and interwoven fiber arrangements of the conical structure effectively dissipate stress and reduce concentration risks, serving as key mechanisms for structural reinforcement​. These findings provide valuable theoretical support for understanding tree branch safety and informing pruning strategies. |
|
Revised lines code |
40) Page15, Section 5. Conclusions |
|
Main concerns: |
26. Line 432 – References – very limited, many pieces of literature from the 1900s |
|
Reply |
This study cites classic works by Shigo and Mattheck, which laid the theoretical foundation for tree biomechanics. Additionally, the revised manuscript incorporates recent literature on finite element analysis and advancements in tree mechanics research to provide more comprehensive reference support. |
|
Revised lines code |
41) Page17, References 1, 21, 22 |
Round 2
Reviewer 1 Report
Comments and Suggestions for Authors
All my previous comments were accepted.
Check, if resinification or heartwood formation are responsible for hardening. The parameters used in model do not belong to acer wood (lines 369-378). Moreover, fig. 10 does not show any acer wood.